# Modified FIB-4 Index in Type 2 Diabetes Mellitus with Steatosis: A Non-Linear Predictive Model for Advanced Hepatic Fibrosis

**DOI:** 10.3390/diagnostics14222500

**Published:** 2024-11-08

**Authors:** Jonghyun Kim, Takanori Ito, Taeang Arai, Masanori Atsukawa, Miwa Kawanaka, Hidenori Toyoda, Takashi Honda, Ming-Lung Yu, Eileen L. Yoon, Dae Won Jun, Kyungjoon Cha, Mindie H. Nguyen

**Affiliations:** 1Research Institute for Convergence of Basic Science, Department of Applied Statistics, College of Natural Sciences, Hanyang University, Seoul 04763, Republic of Korea; dinokimjh@hanyang.ac.kr; 2Department of Gastroenterology and Hepatology, Nagoya University Hospital, Nagoya 466-8560, Japan; tahkun56@gmail.com (T.I.); honda@med.nagoya-u.ac.jp (T.H.); 3Division of Gastroenterology and Hepatology, Nippon Medical School, Tokyo 113-8602, Japan; taeangpark@yahoo.co.jp (T.A.); momogachi@yahoo.co.jp (M.A.); 4Department of General Internal Medicine, Kawasaki Medical School General Medical Center, Okayama 700-8505, Japan; m.kawanaka@med.kawasaki-m.ac.jp; 5Department of Gastroenterology, Ogaki Municipal Hospital, Ogaki 503-8502, Japan; hmtoyoda@spice.ocn.ne.jp; 6Department of Internal Medicine, Kaohsiung Medical University Hospital, Kaohsiung Medical University, Kaohsiung 807, Taiwan; fish6069@gmail.com; 7Department of Internal Medicine, Hanyang University College of Medicine, Seoul 04763, Republic of Korea; mseileen80@hanyang.ac.kr; 8Department of Mathematics, College of Natural Sciences, Hanyang University, 222, Wangsimni-ro, Seongdong-gu, Seoul 04763, Republic of Korea; 9Division of Gastroenterology and Hepatology, Department of Medicine, Stanford University Medical Center, Palo Alto, CA 94304, USA; 10Department of Epidemiology and Population Health, Stanford University Medical Center, Palo Alto, CA 94304, USA

**Keywords:** hepatic fibrosis, diabetes, FIB-4

## Abstract

**Background:** The Fibrosis-4 (FIB-4) index is widely recommended as a first-tier method for screening advanced hepatic fibrosis; however, its diagnostic performance is known to be suboptimal in patients with Type 2 diabetes mellitus (T2DM). We aim to propose a modified FIB-4, using the parameters of the existing FIB-4, tailored specifically for diabetic patients with metabolic dysfunction-associated steatotic liver disease (MASLD). **Methods:** A total of 1503 patients who underwent liver biopsy were divided into T2DM (*n* = 517) and non-T2DM (*n* = 986) groups. The model was developed using multiple regression analysis in the derivation cohort and validated in the validation cohort. Diagnostic accuracy was evaluated using the area under the receiver operating characteristic (AUC) curves. **Results:** Among the 1503 individuals, those with T2DM were older, more likely to be male, and had a higher prevalence of advanced hepatic fibrosis (≥F3) compared to non-T2DM individuals. Independent risk factors for advanced fibrosis in T2DM included age, AST, AST/ALT ratio, albumin, triglycerides, and platelet count. The optimized FIB-4 model for T2DM with MASLD (Diabetes Fibrosis Index) demonstrated superior diagnostic accuracy (AUC 0.771) compared to the FIB-4 (AUC 0.735, *p* = 0.012). The model showed a higher negative predictive value than the original FIB-4 across all age groups in the diabetic group. **Conclusions:** The newly optimized FIB-4 model for T2DM with MASLD (Diabetes Fibrosis Index), incorporating a non-linear predictive model, improves diagnostic performance (AUC) and the negative predictive value in MASLD with T2DM.

## 1. Introduction

Metabolic dysfunction-associated steatotic liver disease (MASLD) is the most common cause of chronic liver disease, affecting approximately one in four individuals in most developed countries [1]. The prevalence of significant hepatic fibrosis in MASLD patients is reported to be around 5.9–8.8% [2,3]. The most important risk factor for the development of hepatic fibrosis is T2DM [4]. The rate of significant hepatic fibrosis in T2DM is higher than in MASLD (5.9~8.8%), estimated to be between 12.5% and 20% [5,6,7]. Early detection and active lifestyle modification for significant or advanced hepatic fibrosis in MASLD is cost-effective and can reduce liver-related events and overall mortality associated with the disease [8,9]. Moreover, with the recent FDA approval of the first treatment for MASLD, early identification and management of significant or advanced hepatic fibrosis in MASLD patients can help reduce the social and economic burden of the disease [9,10].

In relation to the assessment of advanced hepatic fibrosis in T2DM, the American Association for the Study of Liver Diseases (AASLD), the American Diabetes Association (ADA), and the American Gastroenterological Association (AGA) issued recommendations in 2023 for evaluating liver disease in all patients with Type 2 diabetes [11]. Similarly, in 2022, the American Association of Clinical Endocrinology (AACE) released comparable guidelines [12]. Given the high prevalence of MASLD in T2DM, these organizations recommend regular screening for advanced fibrosis in this patient group, even if hepatic steatosis is not clinically evident. This recommendation reflects recent research highlighting the high prevalence of advanced fibrosis in T2DM. The recommended screening methods include an initial risk assessment using the Fibrosis-4 (FIB-4) index and, for at-risk patients, a secondary risk assessment with vibration-controlled transient elastography (VCTE) or the enhanced liver fibrosis (ELF) test [13]. However, the practicality of universal fibrosis screening in T2DM patients remains debatable [14,15,16,17]. Initial screening with the FIB-4 index has limitations, with sensitivity and specificity reported as 73% and 62%, respectively, which may result in missed diagnoses [18]. So, the AGA recommends a second-tier non-invasive test (NIT), such as VCTE, for all T2DM patients, regardless of FIB-4 results, differing from other organizations [19]. Therefore, the current guidelines for screening advanced hepatic fibrosis in T2DM patients have significant unmet needs in both accuracy and practical feasibility [20].

Therefore, there is a need for a new NIT or algorithm to effectively screen for liver fibrosis in patients with T2DM accompanied by hepatic steatosis [21]. To date, many non-invasive serologic tests have been proposed for diagnosing hepatic fibrosis in MASLD patients [22,23]. However, for widespread use in primary care settings, these tests must be cost-effective and free from patent restrictions. For this reason, many clinical guidelines still recommend FIB-4 as the first-tier NIT [24].

This study aims to develop a non-invasive test to identify advanced liver fibrosis in patients with MASLD accompanied by T2DM, confirmed through liver biopsy. We aim to propose a modified FIB-4, using the parameters of the existing FIB-4, tailored specifically for diabetic patients with MASLD.

## 2. Patients and Methods

### 2.1. Study Design

This study was conducted using retrospective liver biopsy data collected from several centers across multiple countries. The study received approval from the Hanyang University Institutional Review Board (HYIRB- 2023-07-063-005). Given the retrospective nature of the research, the requirement for informed consent was waived.

### 2.2. Inclusion and Exclusion Criteria

The study included adults aged 19 years or older who had been diagnosed with T2DM and hepatic steatosis. The diagnosis of hepatic steatosis was confirmed through liver biopsy. Individuals were excluded from the study if their blood test results were insufficient or if their history of alcohol consumption was unclear. We diagnosed NAFLD based on the AASLD guidelines. The inclusion criteria are as follows: (1) the presence of hepatic steatosis was observed through liver biopsy with at least one cardiometabolic risk factor (obesity, impaired fasting glucose, hypertension, hypertriglyceridemia, low HDL cholesterolemia), (2) there must be no excessive alcohol consumption (ethanol intake less than 210 g per week for men and less than 140 g per week for women), and (3) other causes of fatty liver, such as medications, must be excluded. The exclusion criteria are as follows: (1) insufficient data variables, (2) use of medications that can induce fatty liver for more than two weeks, and (3) the presence of chronic liver disease causes such as viral infections (hepatitis C or B virus), primary biliary cholangitis, or autoimmune hepatitis must be excluded.

### 2.3. Definition

MASLD was defined as the absence of significant alcohol consumption (defined as >140 g per week for women and >210 g per week for men) in the past two years. Patients with chronic viral hepatitis, such as hepatitis B or C, were excluded from the study. Additionally, those who were taking medications known to induce hepatic steatosis were not included. Diabetes was defined by the use of diabetes-related medications, a fasting blood glucose level of 126 mg/dL or higher, or a hemoglobin A1c (HbA1C) level greater than 6.5%.

### 2.4. Liver Histology

Liver biopsy samples were scored using the non-alcoholic fatty liver disease activity score (NAS) system, which assigns separate scores for steatosis (0–3), hepatocellular ballooning (0–2), and lobular inflammation (0–3). The fibrosis stage was classified using Brunt’s pathological grading system, ranging from stage 0 to stage 4, with stage 4 being defined as cirrhosis.

### 2.5. In-Depth Analysis of FIB-4 Variables

We investigated the impact of the clinical parameters that constitute FIB-4 on advanced hepatic fibrosis in MASLD patients with T2DM. The interactions between the variables that make up FIB-4—age, platelet count, alanine transaminase (ALT), and aspartate aminotransferase (AST)—were analyzed in the presence of advanced hepatic fibrosis.

### 2.6. Statistical Analysis

To identify independent risk factors for advanced hepatic fibrosis in patients with T2DM, a logistic regression analysis was performed along with T2DM patients’ data. A scoring system based on Polynomial Logistic Regression was created using the factors identified as risk factors (*p* < 0.05) in the logistic regression analysis. The variables used in the model include age, platelets, AST, ALT, and BMI. To determine the coefficients of the model, which consists of the quadratic terms of the main factors, a Genetic Algorithm optimization was employed. The coefficients that yielded the highest AUROC for detecting advanced hepatic fibrosis in T2DM patients were obtained: (Diabetes Fibrosis Index) = 1.4013 × Age − 2.9859 × Plt × (1 − 0.00159 × Plt) + 5.8155 × AST × (1 − 0.00365 × AST) − 1.2014 × ALT + 56.7468 × BMI × (1 − 0.01467 × BMI). Using the formula, a cut-off point was selected from the ROC curve, and the diagnostic performance using this cut-off point was compared with FIB-4. All statistical analyses were conducted using R version 4.4.0.

## 3. Results

### 3.1. Baseline Characteristics of Study Population

Among the 1503 individuals who underwent full liver biopsy, 517 were diagnosed with T2DM. Compared to the non-T2DM group, the T2DM group within the total cohort was older, had a higher proportion of males, and exhibited a greater prevalence of advanced hepatic fibrosis (≥F3) (Table 1). The prevalence of advanced hepatic fibrosis increased with age, and the proportion of patients with diabetes rose as the stage of fibrosis advanced (Figure 1). In the T2DM group, the proportion of individuals with advanced hepatic fibrosis (≥F3) was 30.0% (155/517) (Table 2). Patients with T2DM who had advanced hepatic fibrosis (≥F3) were older, had a higher male proportion, and had a higher BMI compared to those in the F0~2 fibrosis group.

### 3.2. Major Clinical Risk Factors and Their Interactions in Advanced Hepatic Fibrosis

To identify the independent risk factors for advanced hepatic fibrosis in patients with T2DM, a logistic regression analysis was performed (Table 3). The results indicated that age, AST, AST/ALT ratio, albumin, triglycerides, and platelet count were independent risk factors for advanced hepatic fibrosis in the T2DM group. In a separate logistic regression analysis conducted on the non-T2DM group, additional factors such as diabetes status, fasting blood glucose, HbA1c, and gender were identified as independent risk factors for advanced hepatic fibrosis. Interactions among the independent risk factors for advanced hepatic fibrosis in T2DM were also assessed. Notably, age and platelet count, AST and ALT, and platelet count and ALT, as well as age and AST, exhibited non-linear two-dimensional interactions concerning advanced hepatic fibrosis (Figure 2).

### 3.3. Optimized FIB-4 Model for T2DM with MASLD for Advanced Hepatic Fibrosis in T2DM

A non-linear predictive model was constructed to account for the interactions among independent risk factors for advanced hepatic fibrosis in patients with T2DM. The model incorporated five variables: age, platelet count, AST, ALT, and BMI. The diagnostic accuracy of this optimized FIB-4 model for T2DM with MASLD, as measured by the area under the curve (AUC), was 0.771, which was significantly higher than the AUC of 0.735 for FIB-4 (*p* = 0.012) (Figure 3). The optimal cut-off value for the optimized FIB-4 in T2DM, termed the Diabetes Fibrosis Index (calculated as 1.4013 × Age − 2.9859 × Plt × (1 − 0.00159 × Plt) + 5.8155 × AST × (1 − 0.00365 × AST) − 1.2014 × ALT + 56.7468 × BMI × (1 − 0.01467 × BMI)), was 715, yielding a sensitivity of 0.85 and a specificity of 0.52. This performance was superior to the FIB-4 model, which had a sensitivity of 0.82 and a specificity of 0.50. Notably, the diagnostic performance of the FIB-4 model significantly declined with age, particularly in diabetic patients aged 65 years and older, where the AUC dropped to 0.68. In contrast, the optimized FIB-4 model for T2DM with MASLD demonstrated improved diagnostic performance in this older age group, with an AUC of 0.72, surpassing that of the FIB-4 (Table 4).

## 4. Discussion

This study proposes a novel Diabetes Fibrosis Index for identifying patients with advanced hepatic fibrosis in those MASLD patients with T2DM. The optimized FIB-4 model for T2DM with MASLD utilizes clinical parameters similar to those used in the widely adopted FIB-4 index, commonly employed in primary care settings. However, it enhances sensitivity and specificity compared to the traditional FIB-4, particularly addressing the issue of decreased sensitivity associated with age, which is a significant limitation of the FIB-4 [25,26,27]. Our study introduces a Diabetes Fibrosis Index for identifying patients with advanced hepatic fibrosis in those with T2DM based on a large cohort of 1503 biopsy-confirmed patients. The original FIB-4 index has a good ability to rule out advanced fibrosis when a cut-off of 1.3 is used [28]. The novel marker in this study is useful because it shows a higher negative predictive value (NPV) than the original FIB-4 across all age groups in the diabetic group. However, the issue is that the positive predictive value (PPV) is low when using the current cut-off point, suggesting that other NITs, including elastography, may be necessary to confirm advanced fibrosis.

FIB-4 is generally recommended as a first-tier screening tool for high-risk groups in patients with MASLD. It has shown relatively good diagnostic performance for advanced hepatic fibrosis, with reported AUCs ranging from 0.75 to 0.83. However, FIB-4′s diagnostic performance is known to be suboptimal in patients under 35 years of age and those over 65, as well as in those with T2DM, where its sensitivity is notably reduced [29]. Studies such as those by Qadri et al. have reported a diagnostic performance of FIB-4 in predicting advanced hepatic fibrosis in MASLD patients with T2DM with a sensitivity of 73% and a specificity of 62% using the 1.30 cut-off [30]. This suggests that using FIB-4 as a first-tier test in this population could result in missing up to 27% of patients with advanced hepatic fibrosis [31]. Consequently, current guidelines acknowledge the lack of robust evidence for the diagnostic performance of FIB-4 in MASLD patients with T2DM and highlight the need for further research [30].

In this study, we focused on using the same biochemical parameters as the original FIB-4, excluding BMI, to identify advanced hepatic fibrosis in T2DM patients. While various NITs have been proposed for hepatic fibrosis across different at-risk groups, including MASLD and T2DM, their application in primary care settings remains limited. This is largely due to the absence of a clear screening algorithm and the significant societal burden associated with additional costly tests or blood work, particularly when effective treatments are lacking. For this reason, despite its limitations, most clinical guidelines continue to recommend FIB-4 as a first-tier test [32,33,34]. Therefore, our research group sought to improve the performance of the FIB-4 test by utilizing a non-linear equation that reflects the non-linear relationship of the biochemical parameters with advanced hepatic fibrosis (Figure 2).

Despite recommendations in all guidelines to screen for advanced hepatic fibrosis in patients with diabetes due to its high prevalence, this practice is not widely implemented in clinical settings [35]. Simulation studies in the U.S. have suggested that applying FIB-4 as a screening test could result in 40% of all diabetic patients requiring annual vibration-controlled transient elastography (VCTE) tests, and 19% needing referral to a hepatologist each year [30].

The strength of this novel score is that it can exclude cases of advanced fibrosis progression in T2DM-associated SLD using only data obtained from physical examinations. While serum NITs unaffected by T2DM, such as ELF and type 4 collagen 7s, exist, they cannot be used in general health examinations. Therefore, this novel marker will be particularly useful in primary care settings. If additional analysis is conducted, it may be beneficial to compare ROC curves by BMI or degree of T2DM, provided there is sufficient data on HbA1c, not just age.

The limitations of our study are as follows: First, these data are based on a cross-sectional study focused on identifying at-risk groups in T2DM. However, it is also important to verify whether the newly proposed modified FIB-4 score provides superior predictive ability for long-term clinical outcomes in T2DM compared to the conventional FIB-4. Moreover, our use of retrospective data in the study has inherent biases in data collection, such as selection bias or incomplete data, which may still exist and could impact the reliability of the study’s findings. Prospective validation is indeed necessary to confirm the model’s clinical utility. Unfortunately, due to the nature of the database, we were unable to compare the predictive ability for long-term clinical outcomes with the conventional FIB-4. Second, although the new model’s NPV has improved, the improvement in PPV is relatively low. Sensitivity, specificity, NPV, and PPV are highly sensitive to changes in the cut-off point. The authors prioritized maintaining sensitivity and NPV above 90% as the primary goal of a screening test, rather than focusing on increasing PPV. In screening tests, sensitivity and NPV are generally prioritized over specificity and PPV, particularly when used as a first-tier screening tool, where minimizing missed high-risk cases and clearly identifying non-urgent cases are essential. For these reasons, the authors selected an optimal cut-off to achieve sensitivity and NPV close to 90%. Third, the presence of T2DM and insulin resistance are known to be significant independent risk factors for the degree of hepatic fibrosis in MASLD. Additionally, the duration of T2DM and the specific antidiabetic medications used are also reported to play important roles in hepatic fibrosis. However, in this study, data on the duration of T2DM and a detailed history of drug usage were not collected.

In conclusion, this study presents a novel, non-linear predictive model for diagnosing advanced hepatic fibrosis in patients with T2DM, demonstrating superior accuracy compared to the traditional FIB-4 index. By utilizing commonly available clinical parameters, the optimized FIB-4 model for T2DM with MASLD addresses the limitations of FIB-4, particularly its reduced sensitivity in older patients. This model has the potential to improve early detection of advanced fibrosis in T2DM, thereby enhancing patient outcomes. Further validation in clinical settings is warranted to confirm its utility and effectiveness.

## Figures and Tables

**Figure 1 diagnostics-14-02500-f001:**
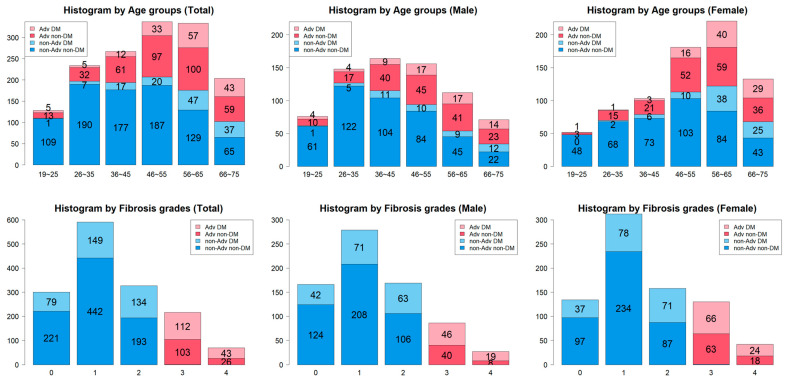
Prevalence of Type 2 diabetes according to fibrosis stage and age.

**Figure 2 diagnostics-14-02500-f002:**
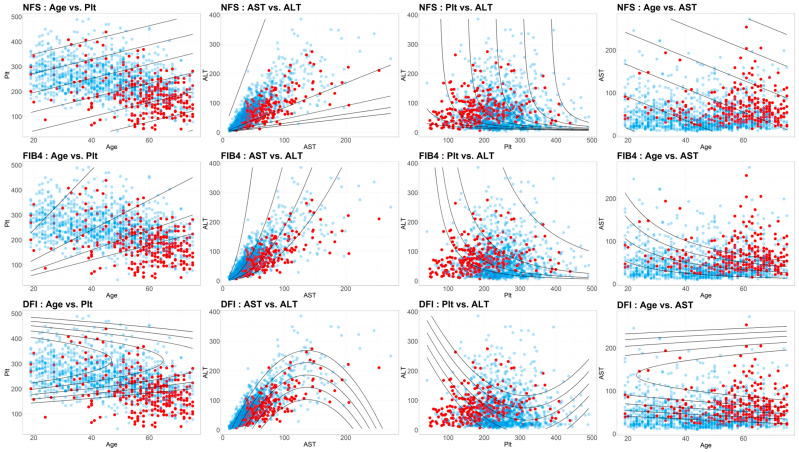
Interaction analysis of major clinical risk factors for advanced hepatic fibrosis.

**Figure 3 diagnostics-14-02500-f003:**
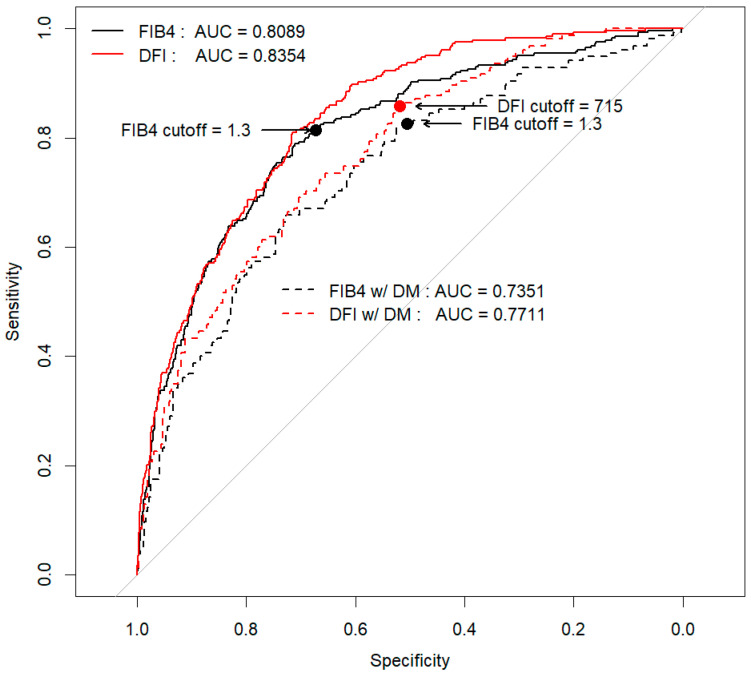
Diagnostic performance of new NIT for T2DM.

**Table 1 diagnostics-14-02500-t001:** Baseline characteristics of total population (T2DM and non-T2DM).

	Total (*n* = 1503)	Non-T2DM (*n* = 986)	T2DM (*n* = 517)	*p*
Age (years) †	48.10 ± 14.85	45.11 ± 14.95	53.81 ± 12.88	<0.001 ^a^
Male	48.4%	49.3%	46.6%	0.352 ^b^
BMI (kg/m^2^) †	29.39 ± 6.59	29.41 ± 7.23	29.355 ± 5.16	0.854 ^a^
Waist circumference (cm) †	95.83 ± 12.46	95.41 ± 11.75	96.50 ± 13.54	0.469 ^a^
Hypertension	42.4%	33.9%	58.6%	<0.001 ^b^
Diabetes	34.4%			
Triglyceride (mg/dL) †	159.46 ± 100.72	156.62 ± 102.69	164.89 ± 96.71	0.129 ^a^
HDL-CROL (mg/dL) †	48.06 ± 14.77	49.00 ± 14.56	46.27 ± 15.00	0.001 ^a^
Cholesterol (mg/dL)	124.39 ± 50.58	127.54 ± 47.84	118.12 ± 55.16	0.004 ^a^
Albumin (mg/dL)	4.33 ± 0.44	4.35 ± 0.42	4.29 ± 0.46	0.007 ^a^
Glucose (mg/dL) †	113.90 ± 35.59	99.45 ± 16.10	141.41 ± 45.07	<0.001 ^a^
AST (IU/L) †	51.54 ± 35.39	47.74 ± 33.87	58.79 ± 37.09	<0.001 ^a^
ALT (IU/L) †	73.63 ± 58.89	71.19 ± 60.20	78.29 ± 56.07	0.023 ^a^
AST/ALT †	0.8697 ± 0.4423	0.8650 ± 0.4634	0.8786 ± 0.3993	0.555 ^a^
Platelets (×10^9^/L) †	236.65 ± 72.67	246.55 ± 70.32	217.78 ± 73.41	<0.001 ^a^
FIB-4 index †	1.58 ± 1.47	1.32 ± 1.24	2.07 ± 1.73	<0.001 ^a^
NAFLD fibrosis score †	−1.82 ± 1.67	−2.46 ± 1.39	−0.58 ± 1.45	<0.001 ^a^
Steatosis	1.39 ± 1.01	1.24 ± 1.03	1.74 ± 0.86	<0.001 ^a^
Inflammation	1.11 ± 0.99	0.91 ± 0.96	1.56 ± 0.90	<0.001 ^a^
Fibrosis	285 (19.0%)	130 (13.2%)	155 (30.0%)	<0.001 ^b^
Prevalence of ≥F2	1218	856	362	
Prevalence of ≥F3	285	130	155	

Data are expressed as number (percent). † Data are shown as mean ± standard deviation. Abbreviations: AST, aspartate transaminase; ALT, alanine transaminase; BMI, body mass index; HDL, high-density lipoprotein; NAFLD, non-alcoholic fatty liver disease; ^a^ Student’s *t*-test; ^b^ Chi-square test.

**Table 2 diagnostics-14-02500-t002:** Baseline characteristics of Type 2 diabetes.

	Total (*n* = 517)	≥F3 (*n* = 155)	F0~2 (*n* = 362)	*p*
Age (years) †	53.8 ± 12.9	57.5 ± 12.2	52.2 ± 12.9	<0.001 ^a^
Male	46.6%	41.9%	48.6%	0.194 ^b^
BMI (kg/m^2^) †	29.36 ± 5.16	29.41 ± 4.48	29.33 ± 5.43	0.865 ^a^
Waist circumference (cm) †	96.50 ± 13.54	98.28 ± 7.75	95.90 ± 14.99	0.266 ^a^
Hypertension	58.6%	65.8%	55.4%	0.036 ^b^
Triglyceride (mg/dL) †	164.89 ± 96.71	141.09 ± 64.46	175.49 ± 106.39	<0.001 ^a^
HDL-CROL (mg/dL) †	46.27 ± 15.00	47.60 ± 17.15	45.68 ± 13.92	0.226 ^a^
Cholesterol (mg/dL)	188.12 ± 55.16	122.30 ± 82.12	116.31 ± 38.01	0.444 ^a^
Albumin (mg/dL)	4.29 ± 0.46	4.20 ± 0.47	4.32 ± 0.45	0.005 ^a^
Glucose (mg/dL) †	141.41 ± 45.07	142.56 ± 46.87	140.91 ± 44.34	0.711 ^a^
AST (IU/L) †	58.79 ± 37.09	65.35 ± 33.84	55.99 ± 38.10	0.006 ^a^
ALT (IU/L) †	78.29 ± 56.08	75.90 ± 47.52	79.31 ± 59.39	0.490 ^a^
AST/ALT †	0.8786 ± 0.3993	0.9893 ± 0.4348	0.8312 ± 0.3738	<0.001 ^a^
Platelets (×10^9^/L) †	217.78 ± 73.41	183.91 ± 73.51	232.279 ± 68.52	<0.001 ^a^
FIB-4 index †	2.07 ± 1.73	3.02 ± 2.20	1.67 ± 1.29	<0.001 ^a^
NAFLD fibrosis score †	−0.58 ± 1.45	0.16 ± 1.56	−0.91 ± 1.27	<0.001 ^a^
Steatosis (mean ± SD)	1.74 ± 0.86	1.77 ± 0.68	1.73 ± 0.92	0.805 ^a^
Inflammation (mean ± SD)	1.56 ± 0.90	1.86 ± 0.80	1.44 ± 0.91	0.006 ^a^
Hepatic fibrosis (%)	30.0%	155	362	
F0/1			228 (63.0%)	
F2			134 (37.0%)	
F3		112 (72.3%)		
F4		43 (27.7%)		

Data are expressed as number (percent). † Data are shown as mean ± standard deviation. Abbreviations: AST, aspartate transaminase; ALT, alanine transaminase; BMI, body mass index; HDL, high-density lipoprotein; NAFLD, non-alcoholic fatty liver disease; ^a^ Student’s *t*-test; ^b^ Chi-square test.

**Table 3 diagnostics-14-02500-t003:** Logistic regression for major clinical risk factors of advanced hepatic fibrosis.

	Total				T2DM			
	Odds Ratio	95% CI Lower	95% CI Upper	*p*	Odds Ratio	95% CI Lower	95% CI Upper	*p*
T2DM	2.799	2.145	3.657	<0.001				
Age (years) †	1.064	1.053	1.076	<0.001	1.063	1.044	1.084	<0.001
Gender (M)	1.518	1.166	1.983	<0.001	1.540	1.020	2.342	0.041
BMI (kg/m^2^) †	0.991	0.970	1.010	0.355	0.991	0.953	1.028	0.636
AST (IU/L) †	1.011	1.008	1.014	<0.001	1.007	1.002	1.013	0.008
ALT (IU/L) †	1.001	0.998	1.003	0.532	0.998	0.994	1.002	0.285
AST/ALT †	2.031	1.539	2.680	<0.001	3.099	1.927	5.071	<0.001
Albumin (mg/dL)	0.381	0.270	0.534	<0.001	0.476	0.274	0.820	0.008
Glucose (mg/dL) †	1.009	1.006	1.012	<0.001	1.006	1.002	1.011	0.004
HBA1c (%)	1.195	1.069	1.340	0.002	1.130	0.979	1.317	0.105
Triglyceride (mg/dL) †	0.999	0.997	1.000	0.111	0.997	0.994	0.999	0.040
HDL-CROL (mg/dL) †	0.997	0.988	1.006	0.532	1.004	0.990	1.016	0.563
LDL-CROL (mg/dL) †	1.000	0.996	1.002	0.770	0.998	0.991	1.004	0.516
Platelets	0.987	0.985	0.989	<0.001	0.987	0.984	0.991	<0.001

Data are expressed as number (percent). † Data are shown as mean ± standard deviation. Abbreviations: AST, aspartate transaminase; ALT, alanine transaminase; BMI, body mass index; HDL, high-density lipoprotein.

**Table 4 diagnostics-14-02500-t004:** Diagnostic performance according to age.

**FIB4**	**Total**							**T2DM**						
**Cut-Off**	**Age Group**	**AUROC**	**Acc**	**Sens**	**Spec**	**PPV**	**NPV**	**Age Group**	**AUROC**	**Acc**	**Sens**	**Spec**	**PPV**	**NPV**
1.3	Total	0.8089	0.6993	0.8134	0.6727	0.3667	0.9393	Total	0.7351	0.6035	0.8258	0.5083	0.4183	0.872
	~25	0.6325	0.9531	0.1667	0.9918	0.5000	0.9603	~25	0.5846	0.7778	0.2000	1.0000	1.0000	0.7647
	26~35	0.6642	0.9231	0.1667	0.9640	0.2000	0.9554	26~35	0.6094	0.7027	0.0000	0.8125	0.0000	0.8387
	36~45	0.6366	0.8427	0.4483	0.8908	0.3333	0.4483	36~45	0.6093	0.7808	0.5000	0.8361	0.3750	0.8947
	46~55	0.7384	0.6469	0.6604	0.6444	0.2574	0.9104	46~55	0.6982	0.6077	0.6970	0.5773	0.3594	0.8485
	56~65	0.7804	0.5285	0.9615	0.3319	0.3953	0.9500	56~65	0.7754	0.5669	0.9649	0.3400	0.4545	0.9444
	66~75	0.7031	0.4608	1.0000	0.1129	0.4211	1.0000	66~75	0.6878	0.4608	1.0000	0.0678	0.4388	1.0000
**New Model**	**Total**							**T2DM**						
**Cut-Off**	**Age Group**	**AUROC**	**Acc**	**Sens**	**Spec**	**PPV**	**NPV**	**Age Group**	**AUROC**	**Acc**	**Sens**	**Spec**	**PPV**	**NPV**
715	Total	0.8354	0.7279	0.8134	0.708	0.3935	0.9421	Total	0.7710	0.6228	0.8581	0.5221	0.4346	0.8957
	~25	0.8757	0.9531	0.3333	0.9836	0.5000	0.9677	~25	0.7538	0.7222	0.4000	0.8462	0.5000	0.7857
	26~35	0.7911	0.8590	0.4167	0.8829	0.1613	0.9655	26~35	0.6656	0.7027	0.6000	0.7188	0.2500	0.9200
	36~45	0.7867	0.7603	0.5517	0.7857	0.2388	0.9350	36~45	0.6995	0.6438	0.5833	0.6557	0.2500	0.8889
	46~55	0.7902	0.7211	0.7170	0.7218	0.3248	0.9318	46~55	0.7423	0.6538	0.7879	0.6082	0.4062	0.8939
	56~65	0.7748	0.6096	0.8942	0.4803	0.4387	0.9091	56~65	0.7912	0.5987	0.9474	0.4000	0.4737	0.9302
	66~75	0.7345	0.5980	0.9625	0.3629	0.4936	0.9375	66~75	0.7217	0.5588	0.9535	0.2712	0.4881	0.8889

## Data Availability

Data in this study include patients’ clinical information, which may be shared upon request to the authors following Institutional Review Board approval.

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
