# Peer review of "Modified FIB-4 Index in Type 2 Diabetes Mellitus with Steatosis: A Non-Linear Predictive Model for Advanced Hepatic Fibrosis"

_diagnostics, 2024, doi:10.3390/diagnostics14222500_

Round 1

Reviewer 1 Report

Comments and Suggestions for Authors

The article Modified FIB-4 Index in Type 2 Diabetes Mellitus with steatosis: A Non-Linear 2 Predictive Model for Advanced Hepatic Fibrosis represents an interesting research in order to create novel index for scoring and early detection of liver fibrosis. Liver steatosis is usually diagnosed by ultrasound, and not by biopsy. 

Please specify if patients whose biopsy samples were taken, are diagnosed steatosis or more severe liver injury. Using novel scoring system it would be great if it can be used to predict fibrosis development in patient with steatosis?

Please discuss this potential impact in prognosis and prediction.

Figures are clear al well presented.

Conclusion is adequate with mentioned limitations of the study.

Author Response

< Reviewer 1 >

The article Modified FIB-4 Index in Type 2 Diabetes Mellitus with steatosis: A Non-Linear 2 Predictive Model for Advanced Hepatic Fibrosis represents an interesting research in order to create novel index for scoring and early detection of liver fibrosis. Liver steatosis is usually diagnosed by ultrasound, and not by biopsy. 

Please specify if patients whose biopsy samples were taken, are diagnosed steatosis or more severe liver injury. Using novel scoring system it would be great if it can be used to predict fibrosis development in patient with steatosis?

  • Response: Thank you for your valuable comment. The newly proposed modified FIB-4 (Diabetes Fibrosis Index) for T2DM with MASLD is specifically designed to screen for advanced hepatic fibrosis. However, the modified FIB-4 (Diabetes Fibrosis Index) has limited ability to predict hepatic steatosis or hepatic inflammation. To predict hepatic steatosis, it may be more appropriate to use established models specifically aimed at hepatic steatosis prediction (ex. Hepatic steatosis index or Fatty liver index).

Please discuss this potential impact in prognosis and prediction.

  • Response: Thank you for this insightful comment. T2DM is associated with a high prevalence of hepatic fibrosis and is one of the most critical independent risk factors for hepatic fibrosis. However, there is a lack of effective blood-based, non-invasive tests to predict hepatic fibrosis in T2DM patients. The commonly used FIB-4 is known to have low sensitivity in T2DM. Therefore, we anticipate that the modified FIB-4 (Diabetes Fibrosis Index) could effectively identify high-risk liver disease groups in T2DM patients, where hepatic fibrosis screening is most crucial. Additionally, this study is a cross-sectional study focused on identifying at-risk groups in T2DM. However, as noted by the reviewer, it is crucial to confirm whether the newly proposed modified FIB-4 score provides superior predictive ability for long-term outcomes in diabetic patients compared to the traditional FIB-4. This would require additional analysis using a longitudinal cohort. Unfortunately, due to the nature of our database, it would indeed be advantageous if the predictive ability for long-term clinical outcomes were included. We have added the following to the discussion as a limitation of this study. Such as following in Discussion part……
  • Discussion (page 14)
    ……… The limitations of our study are as follows. First, this data is based on a cross-sectional study focused on identifying at-risk groups in T2DM. However, it is essential to verify whether the newly proposed modified FIB-4 score provides superior predictive ability for long-term clinical outcomes in T2DM compared to the conventional FIB-4. Unfortunately, due to the nature of the database, we were unable to compare the predictive ability for long-term clinical outcomes with the conventional FIB-4.

Conclusion is adequate with mentioned limitations of the study.

  • Response: Thank you for your comment.

Reviewer 2 Report

Comments and Suggestions for Authors

Kim et al. present the 'Diabetes Fibrosis Index,' a modified FIB-4 index designed to improve the detection of advanced hepatic fibrosis in patients with T2DM and MASLD. Using data from 1,503 biopsy-confirmed patients, the model incorporates non-linear interactions of clinical parameters (age, AST, ALT, platelet count, BMI), achieving higher diagnostic accuracy and negative predictive value (NPV) than the standard FIB-4. Key strengths of the study include its tailored approach for T2DM, use of a large, multi-center cohort, and enhanced diagnostic performance, particularly in high-risk groups. This study is noteworthy and has potential to improve future clinical diagnostics of hepatic fibrosis. However, the reviewer has several comments and concerns regarding this study. The authors should carefully proofread the manuscript.

Major;

  1. Despite improvements in NPV, the model's PPV remains low, suggesting that secondary confirmatory tests (e.g., elastography) are still necessary for diagnosis. Please provide appropriate reasons for this limitation.
  2. Please include data on the duration of T2DM and a detailed history of drug usage.
  3. The non-linear nature of the model may limit its accessibility and ease of implementation in routine clinical settings, particularly in primary care. Please provide appropriate reasons for this.
  4. As a retrospective study, inherent biases in data collection may exist, such as selection bias or incomplete data, which could impact the findings' reliability. Prospective validation is necessary to confirm the model’s clinical utility, as it would allow for standardized data collection, controlled patient selection, and consistent follow-up, thereby providing stronger evidence for the model’s effectiveness and applicability in real-world clinical settings.

Minor;

1.      Abstract; The abstract is difficult to follow. Please revise and include the full forms of FIB-4 and MASLD. T2DM should be Type 2 diabetes mellitus.

2.      IRB; Please provide the approved IRB number.

3.      Figures; All text in the figures is difficult to read. Please enlarge.

4.      Ensure that all abbreviations are provided in the text, such as HBV, HCV, HbA1C.

5.      Line 137. Why (Diabetes Fibrosis Index) is in bold?

6.      BMI should be (kg/m²) not kig/m2

7.      Tables; Please use dL instead of dl.

8.      References; Please use same font. 

Author Response

< Reviewer 2 >

Kim et al. present the 'Diabetes Fibrosis Index,' a modified FIB-4 index designed to improve the detection of advanced hepatic fibrosis in patients with T2DM and MASLD. Using data from 1,503 biopsy-confirmed patients, the model incorporates non-linear interactions of clinical parameters (age, AST, ALT, platelet count, BMI), achieving higher diagnostic accuracy and negative predictive value (NPV) than the standard FIB-4. Key strengths of the study include its tailored approach for T2DM, use of a large, multi-center cohort, and enhanced diagnostic performance, particularly in high-risk groups. This study is noteworthy and has potential to improve future clinical diagnostics of hepatic fibrosis. However, the reviewer has several comments and concerns regarding this study. The authors should carefully proofread the manuscript.

Major;

  1. Despite improvements in NPV, the model's PPV remains low, suggesting that secondary confirmatory tests (e.g., elastography) are still necessary for diagnosis. Please provide appropriate reasons for this limitation.
  • Response: Thank you for this valid and appropriate comment. The modified FIB-4 (Diabetes Fibrosis Index) demonstrates improved diagnostic performance compared to the standard FIB-4. As noted by the reviewer, while the model’s NPV has improved, the improvement in PPV is relatively low due to a conservative cut-off designed to maintain high sensitivity and negative predictive value. Currently, the screening strategy for high-risk groups is based on a two-step algorithm using multiple non-invasive tests (NITs). In general, screening tests prioritize sensitivity and NPV over specificity and PPV, particularly when used as a first-tier tool where it is crucial to minimize missed high-risk cases and to clearly identify non-urgent cases. For these reasons, the authors selected an optimal cut-off that achieves sensitivity and NPV close to 90%. While adjusting the cut-off of the modified FIB-4 (Diabetes Fibrosis Index) could increase PPV, we believe that the current sensitivity of 85.6% and NPV of 90.0% provide a relatively acceptable balance. However, the reviewer’s comment is highly relevant and critical, and we have therefore included this point along with our response as a limitation in the discussion section.
  • Discussion (Page 15)
    Second, although the new model’s NPV has improved, the improvement in PPV is relatively low. Sensitivity, specificity, NPV, and PPV are highly sensitive to changes in the cut-off point. The authors prioritized maintaining sensitivity and NPV above 90% as the primary goal of a screening test, rather than focusing on increasing PPV. This approach aligns with the typical screening strategy for high-risk groups, which is based on a two-step algorithm using multiple NITs. In screening tests, sensitivity and NPV are generally prioritized over specificity and PPV, particularly when used as a first-tier screening tool, where minimizing missed high-risk cases and clearly identifying non-urgent cases are essential. For these reasons, the authors selected an optimal cut-off to achieve sensitivity and NPV close to 90%.

  1. Please include data on the duration of T2DM and a detailed history of drug usage.
  • Response: Thank you for highlighting this crucial point. As the reviewer mentioned, the severity of T2DM could affect the performance of predictive models developed for T2DM. This study is a multinational, multi-center study involving Korea, Japan, and the United States. Unfortunately, our IRB proposals from each institution did not include items related to the duration of T2DM or medication use. Given the practical challenges of revising and re-submitting IRB approvals across more than eight separate hospitals, we were unable to include these data in this revision. However, the authors acknowledge the reviewer’s valid and important point, and we have included this as a limitation in the discussion section as follows:
  • Discussion (Page 15)
    Third, the presence of T2DM and insulin resistance are known to be significant independent risk factors for the degree of hepatic fibrosis in MASLD. Additionally, the duration of T2DM and the specific antidiabetic medications used are also reported to play important roles in hepatic fibrosis. However, in this study, data on the duration of T2DM and a detailed history of drug usage were not collected.

  1. The non-linear nature of the model may limit its accessibility and ease of implementation in routine clinical settings, particularly in primary care. Please provide appropriate reasons for this.
  • Response: Thank you for this valuable comment. As the reviewer mentioned, linear models are generally more intuitive for clinicians to understand and are relatively easier to calculate. However, even with conventional linear models such as the FIB-4 or NAFLD fibrosis score, it is challenging to perform calculations manually in clinical practice; in most cases, these scores are calculated using computer systems. Thus, we believe the accessibility of a non-linear model in clinical settings may not be significantly lower than that of a linear model. Given that FIB-4 is often calculated automatically using specialized software or computer systems in clinical practice, we expect that the clinical applicability of the modified FIB-4 proposed by the authors will not be substantially hindered.

  1. As a retrospective study, inherent biases in data collection may exist, such as selection bias or incomplete data, which could impact the findings' reliability. Prospective validation is necessary to confirm the model’s clinical utility, as it would allow for standardized data collection, controlled patient selection, and consistent follow-up, thereby providing stronger evidence for the model’s effectiveness and applicability in real-world clinical settings.
  • Response: Thank you for this insightful comment. This study attempted to minimize selection bias by collecting data from multiple centers to reduce bias; however, as the reviewer pointed out, the retrospective nature of the study means inherent biases in data collection, such as selection bias or incomplete data, may still exist, which could impact the reliability of the study’s findings. Prospective validation is indeed necessary to confirm the model’s clinical utility. Prospective validation would enable standardized data collection, controlled patient selection, and consistent follow-up, thereby providing stronger evidence for the model’s effectiveness and applicability in real-world clinical settings. We will include this important point raised by the reviewer as a limitation in the discussion section of the study. Authors acknowledge the reviewer’s valid and important point, and we have included this as a limitation in the discussion section as follows:
  • Discussion (Page 15)

…. Moreover, our retrospective data of the study has inherent biases in data collection, such as selection bias or incomplete data, may still exist, which could impact the reliability of the study’s findings. Prospective validation is indeed necessary to confirm the model’s clinical utility.

Minor;

  1. Abstract; The abstract is difficult to follow. Please revise and include the full forms of FIB-4 and MASLD. T2DM should be Type 2 diabetes mellitus.
  • Response: In accordance with the reviewer's suggestion, we will revise the abstract to improve clarity and will include the full terms for FIB-4 and MASLD.

  1. IRB; Please provide the approved IRB number.
  • 저자의 지적 사항에 따라 IRB 번호를 추가 기입하겠습니다.
  • Response: We will add the IRB approval number as requested by the reviewer. (HYIRB- 2023-07-063-005)

  1. Figures; All text in the figures is difficult to read. Please enlarge.
  • Response: We have revised the figures to enhance readability by enlarging the text.

  1. Ensure that all abbreviations are provided in the text, such as HBV, HCV, HbA1C.
  • Response: As suggested by the reviewer, we have ensured that all abbreviations are defined in the text.

  1. Line 137. Why (Diabetes Fibrosis Index) is in bold?
  • Response: The bold formatting of "Diabetes Fibrosis Index" in Line 137 was an error. We have corrected it.

  1. BMI should be (kg/m²) not kg/m2
  • Response: We have corrected the unit for BMI to the appropriate format (kg/m²).

  1. Tables; Please use dL instead of dl.
  • Response: We have updated the units in the tables to use "dL" as requested.

  1. References; Please use same font. 
  • Response: We have revised the references to ensure consistent font usage.

Round 2

Reviewer 2 Report

Comments and Suggestions for Authors

The authors successfully addressed all of the reviewers' concerns and questions.